# TIME- AND LABEL-EFFICIENT ACTIVE LEARNING BY DIVERSITY AND UNCERTAINTY OF PROBABILITIES

## ABSTRACT

We propose FALCUN, a novel deep batch active learning method that is label- and time-efficient. Our proposed acquisition uses a natural, self-adjusting balance of uncertainty and diversity: It slowly transitions from emphasizing uncertain instances at the decision boundary to emphasizing batch diversity. In contrast, established deep active learning methods often have a fixed weighting of uncertainty and diversity. Moreover, most methods demand intensive search through a deep neural network's high-dimensional latent embedding space. This leads to high acquisition times during which experts are idle as they wait for the next batch to label. We overcome this structural problem by exclusively operating on the low-dimensional probability space, yielding much faster acquisition times. In extensive experiments, we show FALCUNs suitability for diverse use cases, including image and tabular data. Compared to state-of-the-art methods like BADGE, CLUE, and AlfaMix, FALCUN consistently excels in quality and speed: while FALCUN is among the fastest methods, it has the highest average label efficiency.

## 1 INTRODUCTION

Deep neural networks have proven their worth in various fields and are widely used for solving complex tasks. Their great success depends largely on the availability of labeled data. However, while large volumes of unlabeled data are often easily accessible, the labeling process is time-consuming and costly. Especially for medical or industrial applications, ground truths are rare and annotations are particularly expensive due to the need for domain experts. Active learning (AL) approaches reduce annotation efforts by iteratively selecting and labeling the most valuable instances that contribute most to improving performance. Due to the computational overhead of training neural networks, most DAL methods do not update the model after every single query but acquire entire batches of instances for annotation at once (Ash et al., 2020). This batch setting introduces new challenges: How can we select the most informative instances while minimizing the information overlap? E.g., only selecting uncertain instances might result in very similar, redundant information, while simply maximizing diversity might result in labeling uninformative samples.

To measure diversity, established AL methods often exploit the latent representations of a deep learning model (Ash et al., 2020; Sener & Savarese, 2018; Prabhu et al., 2021; Zhdanov, 2019). The learned features are used to obtain similarities between instances and subsequently for diversity

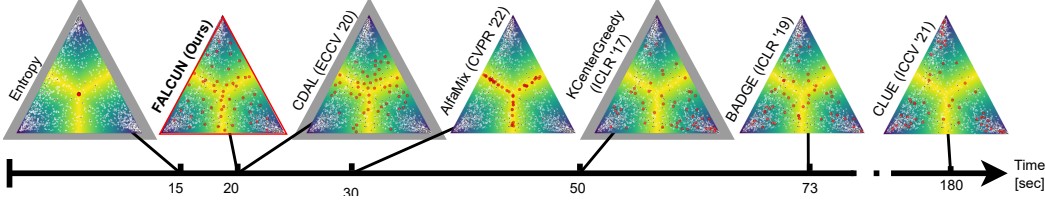

Figure 1: Each simplex illustrates the probability space of a three-class subset of MNIST. Maximum probabilities of each class are in the corners as also reflected by a darker color. Small black and white dots are objects in $\mathcal{L}$ and $\mathcal{U}$, respectively. Larger red dots indicate instances selected by an AL method. FALCUN acquires objects very fast and returns a meaningful selection.

selection. While the learned internal features of the model may eventually provide good discrimination between inputs, they can be very high dimensional depending on the domain and commonly used models. E.g., the dimensionality of the last hidden layer for commonly used architectures (see Kirsch et al. (2019); Ash et al. (2020); Parvaneh et al. (2022)) is 512 in ResNet18, 2048 in ResNet50, and 4096 in VGG16. Thus, searching the feature space can be very time-intensive, leading to acquisition times of up to several days. However, especially for the main use cases of AL, where annotation costs should be reduced, it is very costly to have professionals arrive on multiple days instead of only once in the best case. Imagine an expert visiting an institution for only a short time or a model that needs to be trained until an imminent deadline. Unnecessarily long computation times are also prohibitive from an ecological point of view. **We aim at a fast method that has reliably low runtimes and high quality independently of the model complexity.**

*Diversity* and *uncertainty* are the two main ingredients for effective AL methods. However, *how* to best combine both is an open research question. Current approaches often treat the latent and probability spaces separately (Parvaneh et al., 2022; Prabhu et al., 2021), requiring an additional step to merge the extracted information into a coherent acquisition. One drawback of such handcrafted combinations is the risk of overemphasizing either uncertainty or diversity, resulting in inconsistent label efficiency across various datasets. Furthermore, these combinations may rely on additional parameters (Zhdanov, 2019), that are hard to select in advance. Others try to unify all information into one large representation (e.g.,Ash et al. (2020)). While this enables simultaneous uncertain and diverse selection, it increases the acquisition time. **We aim at a natural combination of uncertainty and diversity without complicated merging processes or threshold parameters.**

Our new AL approach called FALCUN (**F**ast **A**ctive **L**earning by **C**ontrastive **UN**certainty) queries instances that lead to high-quality results for deep learning (DL) while simultaneously being faster than comparative methods. We achieve that by using the output probabilities of the DL model as a low-dimensional representation to determine the similarity between input samples. FALCUN's acquisition is largely independent of the model architecture and depends only on the number of output classes, which is usually much smaller than the latent representations. Figure 1 shows the runtimes and selected samples of different AL methods for a subset of the MNIST dataset with three classes: our approach selects diverse and informative samples while it is among the fastest. We discuss this figure in more detail in Section 3. The main benefits of FALCUN are:

- Label Efficiency: Across varying datasets, active learning settings and model architectures FALCUN is always one of the most label-efficient methods.
- Speed: Among competitors reaching similar accuracy, FALCUN is the **fastest**.
- Scalability, flexibility, and robustness: FALCUN's speed and quality are **independent of the model architecture** and complexity as it does not rely on high-dimensional latent representations.
- Diversity: Even on **high-redundancy data sets**, FALCUN finds a **diverse** set of instances.
- Explainability and simplicity: FALCUN is **easy** to understand and implement, as we use the output probabilities instead of the latent ("black box") representations. Our code is available under `https://anonymous.4open.science/r/falcun-F1C1`.

## 2 METHODOLOGY OF FALCUN

### 2.1 NOTATION

Our task is multi-class classification on an input space $\mathcal{X}$ and a set of labels $\mathcal{Y} = \{1, \ldots, C\}$ for $C$ classes. We consider pool-based AL, where a small initial labeled set $\mathcal{L} \subset \mathcal{X}$ is uniformly drawn from the unlabeled data distribution. The remaining data objects belong to the unlabeled set $\mathcal{U} = \mathcal{X} \setminus \mathcal{L}$ of size $N$. At each AL round, $Q$ samples are selected for annotation and retraining of the model. A classification model $f(x; \theta) \to \mathbb{R}^C$ with parameters $\theta$ maps a given input $x \in \mathcal{X}$ to a $C$-dimensional vector. Correspondingly, $f(x; \theta^{-1}) \to \mathbb{R}^D$ denotes the $D$-dimensional latent representation w.r.t. the penultimate layer of the classifier. The softmax function applied on the model output given by $f(x; \theta)$ for an object $x$ returns the output probability vector $\mathbf{p}(x) \in [0, 1]^C$. We measure the distances between two instances $x_1$ and $x_2$ based on their probabilities using the L1 norm $|| \cdot ||_1$: $dist(\mathbf{p}(x_1), \mathbf{p}(x_2)) := ||\mathbf{p}(x_1) - \mathbf{p}(x_2)||_1 = \sum_{i=1}^{C} |p_i(x_1) - p_i(x_2)|$.

---

**Algorithm 1** Our AL Algorithm FALCUN

**Input**: Unlabeled data pool $\mathcal{U}$, initially labeled data pool $\mathcal{L}$, number of acquisition rounds $R$, query-size $Q$, model $f(x; \theta)$, relevance factor $\gamma$

1: Train initial weights $\theta_0$ on $\mathcal{L}$ by minimizing $\mathbb{E}_{\mathcal{L}}[l_{ce}(f(x; \theta), y)]$
2: **for** $r = 1, 2, \ldots, R$ **do**
3:     Initialize empty query set: $\mathcal{Q} = \{\}$
4:     $\forall x \in \mathcal{U}$ : Compute class probabilities $\mathbf{p}(x)$
5:     $\forall x \in \mathcal{U}$ : Initialize uncertainty $u(x)$ and diversity $d(x)$ scores with Equations (1) and (2)
6:     **for** $q = 1, \ldots, Q$ **do**
7:         $\forall x \in \mathcal{U}$ : Calculate relevance score $r(x)$ with Equation (5)
8:         Sample from distribution with probability proportional to relevance using Equation (6)
9:         $\mathcal{Q} = \mathcal{Q} \cup x_q$
10:        $\forall x \in \mathcal{U}$ : Update diversity values $d(x)$ using Equations (3) and (4)
11:    **end for**
12:    Receive new labels from oracle for instances in $\mathcal{Q}$
13:    $\mathcal{L} = \mathcal{L} \cup \mathcal{Q}$
14:    $\mathcal{U} = \mathcal{U} \setminus \mathcal{Q}$
15:    Train new model $\theta_r$ from scratch on $\mathcal{L}$ by minimizing $\mathbb{E}_{\mathcal{L}}[l_{ce}(f(x; \theta), y)]$
16: **end for**
17: **return** Final parameters $\theta_R$ obtained in round $R$

---

## 2.2 ACQUISITION

In contrast to many previous methods, FALCUN only operates on the model's output probabilities. Instead of employing two independent aspects exploiting the latent space for diversity and the probability space for uncertainty, FALCUN directly uses the probability representation to select diverse *and* uncertain instances.

**Uncertainty Component** For uncertainty, we use the margin uncertainty, i.e., the difference between the probabilities of its two most probable classes:

$$u(x) := 1 - (\mathbf{p}(x)[c_1] - \mathbf{p}(x)[c_2]), \tag{1}$$

where $0 \leq u(x) \leq 1$. Margin is a common choice for uncertainty (Roth & Small, 2006; Bahri et al., 2022; Jiang & Gupta, 2021) and naturally captures the class boundaries between each class pairs. In contrast to other simple uncertainty scores like entropy or least confidence, the margin uncertainty has an extremal function that contains a diverse set of samples: its optima lie on the class boundaries in the probability space. Hence, it emphasizes diverse regions to be of equal interest and naturally captures more dissimilar concepts.

**Diversity Component** A common way to incorporate diversity is to favor instances with large distance to the current batch of instances. However, a good initialization for the diversity is hard to measure when the query batch is still empty. Some methods measure the distance to the already labeled instances, but a problem is that this again requires vast computations depending on the size of the labeled set and does not necessarily reflect the difference between hard-to-learn concepts vs easy-to-learn concepts. A good starting point is to target instances that contain highly contrasting concepts to those that are already easy to classify. (i.e., the corners of the simplex with 100% prediction probability for one class).

However, e.g., simply maximizing the distance to the one-hot, perfect prediction would activate the most central place in the probability simplex in a convex way as there is a global optimum (where all classes are equally probable). This would initially lead to a low diversity, as a group of similar instances would be chosen instead of instances of diverse concepts. Thus, an optimization function for diverse samples should not have a global optimum. As previously mentioned, the *margin* uncertainty $u(x)$ fulfills this property. It incorporates the distance from the best and second best prediction. In Appendix A.2, we show the relation of margin uncertainty and the distances of objects to the one-hot encodings of their two most probable classes in the probability space. Because

of that strong relation, we initialize the diversity score with the already calculated margin uncertainty and update it with every chosen sample $x_q$:

$$d'_{init}(x) := u(x) \quad (2) \qquad d'(x) \leftarrow \min(d'(x), dist(\mathbf{p}(x), \mathbf{p}(x_q))) \quad (3)$$

Finally, we normalize the values to $[0, 1]$ to align it with the uncertainty scores:

$$d(x) := \frac{d'(x) - \min_{x \in X}(d'(x))}{\max_{x \in X}(d'(x)) - \min_{x \in X}(d'(x))}. \tag{4}$$

**Final Relevance Score** For every point $x$, we calculate a relevance score $0 \leq r(x) \leq 2$, which changes over the course of each AL round. We combine the uncertainty and the diversity component by defining $r(x)$ as the sum of the uncertainty $u(x)$ and the normalized adaptive diversity score $d(x)$:

$$r(x) := u(x) + d(x). \tag{5}$$

Note that the values in $u(x)$ are static within one acquisition, but the diversity scores $d(x)$ are updated with every chosen query instance. Thus, the diversity slightly overshadows when the regions with highest uncertainty are exhausted. When there is decent coverage in the probability space and diversity scores denote a uniform distribution, the focus is more on uncertainty. Hence, there is always a natural balance between uncertain and diverse selection depending on the current query batch. Given the relevance scores, we choose $x$ as a next query sample $x_q$ with probability

$$x_q \sim \frac{r(x)^\gamma}{\sum_{x \in \mathcal{U}} r(x)^\gamma}, \tag{6}$$

where $\gamma$ is a parameter that controls the influence of the relevance scores. Note that $\gamma = 0$ corresponds to a uniform selection and larger values for $\gamma$ result in a stronger focus on the calculated relevance scores getting more and more deterministic (rich values get richer). Thus, $\gamma$ controls the trade-off between exploration (more randomness) and exploitation (more focus on larger values in $r(x)$). We analyze the effect of $\gamma$ in Section 4.4.

One acquisition round stops when the batch $\mathcal{Q}$ contains $B$ samples and returns the query batch $\mathcal{Q}$, which will be sent to the oracle for annotation. The pseudo-code for FALCUN's acquisition function and the superordinate AL loop with all steps is shown in Algorithm 1.

## 3 RELATED WORK

We give an overview of the most important state-of-the-art methods, which also serve as our comparative methods in Section 4. A visual comparison including runtimes is given in Figure 1.

Querying the most *uncertain* instances is a key concept in AL. It indicates high *informativeness* such that the model can effectively refine the decision boundary and enhance generalization performance if included in the training (Settles, 2009). To reduce training times, it is common to re-train only after multiple instances were sent to the oracle in batches instead of after each single annotation. This batch-setting poses additional challenges: To minimize the information overlap within a batch, *diversity* is an important criterion in addition to informativeness. E.g., in Figure 1, Entropy (Wang & Shang, 2014) as a non-diversity aware method selects only duplicates.

Diversity-based approaches (e.g.,Sener & Savarese (2018); Sinha et al. (2019); Kirsch et al. (2019)) focus on minimizing the redundancy in the query-batch. An early and prominent representative is KCENTERGREEDY (Sener & Savarese, 2018). However, it only focuses on coverage, which can lead to the selection of instances that are not improving the model. E.g., in Figure 1, KCenterGreedy focuses on the corners of the triangle where the model's prediction already has a very high certainty.

To overcome the challenges of solely uncertainty or diversity-based methods, *hybrid* approaches (Kirsch et al., 2019; Ash et al., 2020; Prabhu et al., 2021) combine both paradigms. However, how to best combine uncertainty and diversity is an ongoing challenge. Similarly to KCenter-Greedy, many methods perform a thorough search on the latent features to determine the diversity between instances. E.g., BADGE (Ash et al., 2020) performs $k$-Means++ sampling on so-called gradient embeddings where large gradients are an indicator of uncertainty. However, these gradient embeddings can have high dimensionalities as they depend on the number of classes times the

hidden dimensionality of the penultimate layer and get very high-dimensional. Thus, the distance calculation is computationally expensive. Other methods perform weighted $k$-Means clustering on the latent representations (Prabhu et al., 2021; Zhdanov, 2019) where weights are some kind of uncertainty estimate and select the most central point from each cluster for annotation. Due to the repeated clustering, these methods are also computationally expensive. AlfaMix (Parvaneh et al., 2022) also performs k-means clustering on latent representations. In contrast to other methods, AlfaMix only clusters on a candidate pool determined by interpolating features in the latent space. The distance calculation is not performed on the whole unlabeled pool, which increases the computational efficiency. However, as shown in Figure 1, AlfaMix oversamples the decision boundary. This might lead to the selection of redundant instances, especially for highly repetitive datasets. CDAL (Agarwal et al., 2020) uses a similar approach as KCenterGreedy but works on the output probabilities. It selects instances where the predicted probability is furthest away from already labeled instances. However, a problem is that some concepts in the data might be harder to learn than others. If instances get labeled, but the model needs more information in such a region, CDAL would not choose instances in the region. Task-specific hard-to-learn concepts might be ignored.

Note that all mentioned and tested methods are either slower than FALCUN or yield less label-efficient results. Results in Figure 1 that are surrounded by a gray triangle (Entropy, CDAL, KCenterGreedy) are less label-efficient than FALCUN in 31-51% of all experiments (and worse than random sampling in 13-31%). Methods yielding similar label-efficiency as FALCUN (BADGE, CLUE) have acquisition times that are multiple times larger than FALCUN's. Even the most recent method, AlfaMix, has grave disadvantages compared to FALCUN: it oversamples at the decision boundary leading to the selection of many duplicates in highly redundant datasets.

## 4 EXPERIMENTS

We evaluate the effectiveness of established AL methods and FALCUN regarding quality and acquisition runtime in isolation as well as in combination to get a complete picture. We use a broad range of datasets including grayscale images (MNIST (LeCun et al., 1998), FashionMNIST (Xiao et al., 2017), and EMNIST), colored images (SVHN (Netzer et al., 2011), BloodMNIST, DermaMNIST (Yang et al., 2023)), and tabular datasets from the Openml benchmark [1] suite (Ids: 6, 155, 156). We further include challenging versions of MNIST containing duplicate images, similar to RepeatedMNIST proposed in Kirsch et al. (2019). For this, we randomly keep 20%, 10%, and 5% unique original images and fill the rest with duplicated versions with added Gaussian noise. For the grayscale images, we use LeNet as a model architecture and use 20 epochs without early stopping. Due to the higher complexity of the colored data, we perform experiments using pre-trained weights and initializing the model with the weights from the previous round as proposed in Parvaneh et al. (2022). We include LeNet and Resnet18 for classification. We train 100 epochs with early stopping when training accuracy reaches 99% following the setting of Ash et al. (2020). We use a learning rate of 0.001 and an Adam optimizer. For the tabular data sets, we use a simple multi-layer-perceptron (MLP) with two layers as proposed in Ash et al. (2020) with a hidden dimensionality of 1024. The learning rate is 0.0001 with an Adam optimizer. We also use early stopping when a training accuracy of 99% is reached. We do not use weight decay or a learning rate scheduler. We perform all experiments with different seeds five times and ten acquisition rounds. We compare to state-of-the-art methods using diversity and uncertainty: BADGE (Ash et al., 2020), CDAL (Agarwal et al., 2020), CLUE (Prabhu et al., 2021), and ALFAMIX (Parvaneh et al., 2022). We also include a diversity baseline: KCENTERGREEDY (Sener & Savarese, 2018), an uncertainty baseline: ENTROPY sampling (Settles, 2009), and the passive baseline RANDOM sampling. For FALCUN, we set $\gamma = 10$ as elaborated in Section 4.4.

### 4.1 LABEL EFFICIENCY

A major goal of AL is to minimize the number of samples that experts need to label to obtain an as good as possible model. Figure 2 shows the learning curves of diverse architectures and query sizes for the evaluated datasets. The x-axis depicts the labeling budget and the y-axis gives the average accuracy for varying AL methods. We see that FALCUN is among the best-performing methods for varying query sizes, data types, and model architectures.

---

[1] https://www.openml.org/

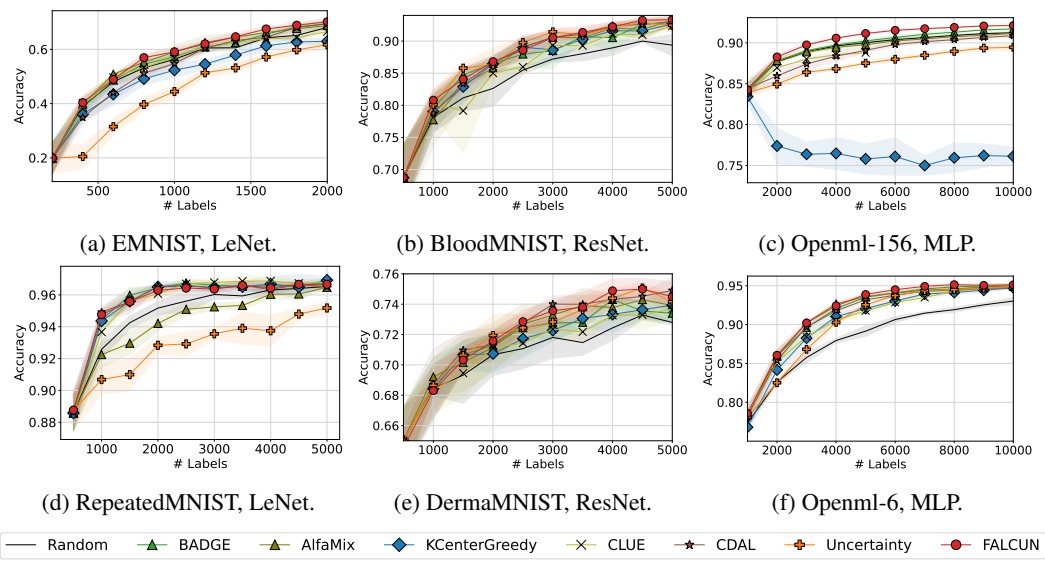

(a) EMNIST, LeNet.  (b) BloodMNIST, ResNet.  (c) Openml-156, MLP.

(d) RepeatedMNIST, LeNet.  (e) DermaMNIST, ResNet.  (f) Openml-6, MLP.

Figure 2: Average test accuracy vs labeling budget for all active learning methods evaluated on greyscale (a,d), RGB (b, e) and tabular data (c, f).

FALCUN is not only a good choice for image data, but also yields the strongest results on the tabular data: in contrast to all other competitors, it consistently outperforms random sampling on the Openml-156 dataset. Note that the ranking of the best-performing methods is not the same over varying settings. E.g., Entropy, an only uncertainty-based technique, yields good results on the bloodmnist dataset, but underperforms on certain other datasets such as EMNIST, RepeatedMNIST or Openml-156. In contrast, KCenterGreedy, a solely diversity-based approach, only yields fairly good results on the highly redundant dataset RepeatedMNIST, but is performing poorly on Openml-156. Not surprisingly, some datasets and settings benefit more from uncertainty

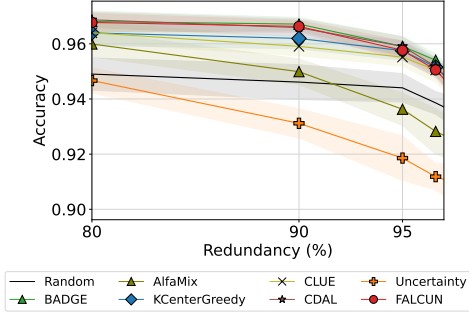

Figure 3: Final average test accuracy for varying redundancy ratios.

and others might work better with diversity. However, in a real-world scenario, it is hard to predict which strategy will work best for a certain use case. Hence, a good active learning strategy has to be successful on a broad range of settings.

FALCUN yields strong results on *all* datasets. We especially want to emphasize that though only operating on the output probabilities, FALCUN's success is not diminished on RepeatedMNIST. Figure 3 shows how the performance of all AL methods drops for varying redundancy ratios of the RepeatedMNIST dataset. Besides Entropy sampling, AlfaMix's quality decreases heavily for high redundancy as it only samples from a certain region, which leads to queries with high similarity (see also Figure 1). All learning curves of all conducted experiments can be seen in the Appendix.

**Dueling Matrix Over All Experiments** In AL, it is hard to compare all learning curves from all experiments, and sometimes a clear winner is hard to find. Hence, similar to previous works (Ash et al., 2020; Parvaneh et al., 2022), we provide the dueling matrix for a comprehensive analysis of the methods' performance over all experimental settings. The column-wise entries in the matrix in Figure 4 show the amount of **losses**, and the row-wise entries indicate the amount of **wins** against each other method (in %). A win means that for a specific experimental setting, i.e., a specific dataset, acquisition round, query size, and model architecture, comparing the results of 5 runs, a method has statistically better accuracy than the other method (with p-value=0.05). A loss is defined analogously.

Losses and wins do not necessarily sum up to 100% as two methods can perform comparably well with no statistical difference. When discussing the quality of an AL method it is hence important to evaluate the wins as well as the losses. The bottom row and the rightmost column denote the average losses and wins over all experiments compared to all other AL methods. Designing a robust method is hard when the characteristics of a dataset are unknown in advance. E.g., some datasets benefit more from diversity (e.g., RepeatedMNIST), some work well with plain uncertainty (e.g., SVHN), some do not benefit from the active acquisition in general, and others work best with specific combinations. Thus, methods with a suboptimal combination of uncertainty and diversity or such focusing on only one direction

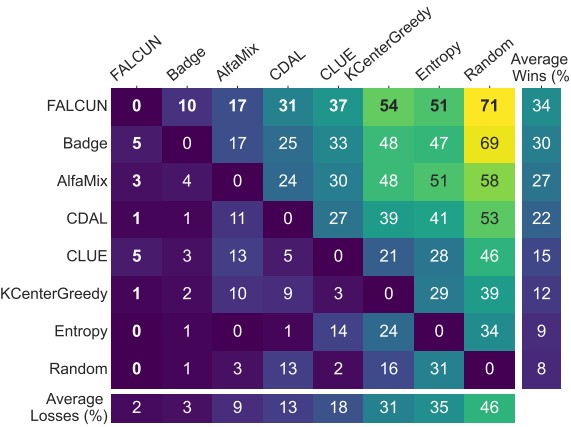

Figure 4: Duelling matrix: The last column gives the percentage of wins of the respective method. The last row gives the percentage of losses.

only succeed in particular tasks. In contrast, FALCUN is consistently strong over a wide range of datasets, as the dueling matrix in Figure 4 shows. FALCUN has the most wins (highest numbers in every column) compared to every other method and also the most wins over random sampling. Simultaneously, it has the fewest losses. Besides FALCUN, only BADGE is *never worse than random sampling*, which is one of the most important criteria for successful AL methods.

Table 1: Time complexity of acquisitions w.r.t. query size $Q$, unlabeled pool size $N_u$, number of classes $C$, size of the labeled pool $N_l$, number of cluster rounds $i$, and a method-specific candidate pool in AlfaMix $N_{cp}$, which is smaller than $N_u$.

| Algorithm | Time Complexity |
|---|---|
| CLUE | $\mathcal{O}(Q \cdot N_u \cdot i \cdot D)$ |
| KCenterGreedy | $\mathcal{O}(N_l \cdot N_u \cdot D + Q \cdot N_u)$ |
| CDAL | $\mathcal{O}(N_l \cdot N_u \cdot C + Q \cdot N_u)$ |
| BADGE | $\mathcal{O}(Q \cdot N_u \cdot C \cdot D)$ |
| AlfaMix | $\mathcal{O}(Q \cdot N_{cp} \cdot i \cdot D)$ |
| **FALCUN (Ours)** | $\mathcal{O}(Q \cdot N_u \cdot C)$ |

Table 2: Min. and max. average cumulated query time (after the whole querying budget is exhausted) among all experiments within the scalability analysis for all methods.

| | min | max |
|---|---|---|
| Entropy | 1.8 sec | 21 min |
| CDAL | 1 min | 80 min |
| **FALCUN (Ours)** | 1.5 min | 97 min |
| AlfaMix | 7.3 min | 175 min |
| KCenterGreedy | 11.8 min | 25 hrs |
| BADGE | 31.5 min | 208 hrs |
| CLUE | 92 min | >227 hrs |

## 4.2 Query Time Efficiency

The training for the grayscale image datasets and tabular datasets is arguably fast (around 1 minute for the last AL round). For the colored image data, training takes around 75 minutes in the last round. In such situations, the limiting factor for the overall runtime is the query time. We show the runtime complexities in Table 1. Note that the runtime complexity of our acquisition is dependent on the size of the unlabeled pool, the query size and the number of classes ($\mathcal{O}(Q \cdot N_u \cdot C)$) but not on the hidden dimensionality $D$. BADGE, one of the strongest competitors regarding label efficiency, has a worse runtime complexity with $\mathcal{O}(Q \cdot N_u \cdot C) \in \mathcal{O}(Q \cdot N_u \cdot C \cdot D)$. That leads to multiple times higher run times compared to FALCUN as shown in Table 2.

We systematically analyzed the scalability of all tested methods by varying dataset size, query size, and hidden dimensionality of the multilayer perceptron evaluated for the largest of all datasets (i.e., Openml-156) and report the results in Figure 18 in Appendix D. Table 2 summarizes these extensive experiments by giving the smallest and largest average query times among all of those experiments. We stopped each experiment after 10 days (e.g. CLUE). CDAL, followed closely by FALCUN, is the fastest among all tested methods. In the fastest setting, when the unlabeled pool contains 20,000 objects, FALCUN is only half a minute slower than CDAL. In the most challenging setting with a latent dimension of 4096, FALCUN is only 17% slower.

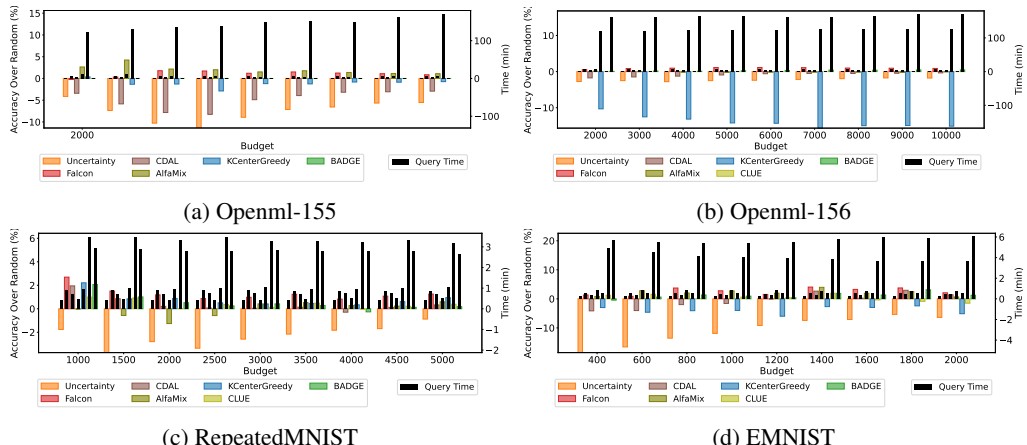

Figure 5: Runtimes (black bars, smaller is better) and improvement over random sampling in average test accuracies (colored bars, larger is better) for all acquisition rounds for tabular data (Openml-155 and Openml-156) and grayscale data (RepeatedMNIST, EMNIST).

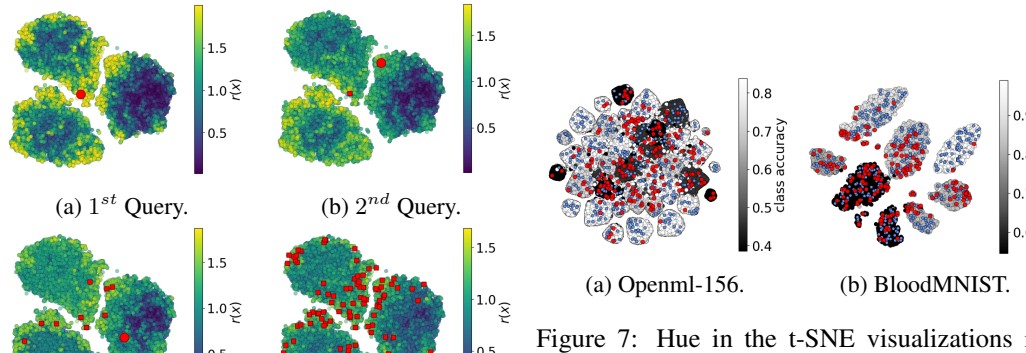

Figure 6: Exemplary course of relevance scores $r(x)$ and their dependency of selected queries (red) on 3-class MNIST, t-SNE visualization.

Figure 7: Hue in the t-SNE visualizations indicates the predictive accuracy of the model on the respective class. Initially sampled objects are blue, samples chosen by FALCUN in the first acquisition round are red. FALCUN selects diverse instances favoring classes that are harder to distinguish by the current model: "darker" classes contain more red dots.

Regarding quality and runtime together, Figure 5 shows the improvement over random sampling in terms of average accuracy per method (colored bars) and the corresponding query time in minutes in a certain acquisition round (black thin bars) for all tested methods. *Large accuracy bars are better* whereas *smaller time bars are better*. FALCUN (red bars) has strong performance on all datasets and never has worse average accuracy than random sampling (i.e., values smaller than zero). CLUE and especially BADGE perform on par in some settings, but their query times are much higher, in some cases up to $> 200$ hours. AlfaMix is fast and has good quality on Openml-155 and decent performance on EMNIST. However, AlfaMix is prone to duplicates: it performs even worse than random sampling on RepeatedMNIST in all acquisition rounds. CDAL is quite fast but performs worse than random sampling more often, especially for small budgets on EMNIST and both tabular datasets. It is worse than FALCUN on all four datasets. Entropy is fast, but not label-efficient. KCenterGreedy is fast for smaller datasets (e.g., RepeatedMNIST and EMNIST) but does not scale well to larger datasets (see Openml-156) and is only comparably label-efficient for the redundant dataset RepeatedMNIST because it has the strongest emphasis on maximizing diversity. FALCUN has a robust performance across all datasets and low query times (never above 10 minutes).

Table 3: Ablation Study: Average test accuracies over 5 runs for varying $\gamma$ and deterministic sampling ("det.") with and without diversity ($d(x)$) and uncertainty ($u(x)$). Red numbers indicate worst and bold numbers indicate best accuracy in a column. Note that $\gamma = 0$ equals random sampling.

| $\gamma$ | $u(x)$ | $d(x)$ | EMNIST | | | RepeatedMNIST (90%) | | | BloodMNIST | | | Openml-6 | | |
|---|---|---|---|---|---|---|---|---|---|---|---|---|---|---|
| | | | 400 | 1000 | 2000 | 1000 | 2500 | 5000 | 1000 | 2500 | 5000 | 2000 | 5000 | 10000 |
| 0 | - | - | 39.00 | 56.03 | 67.64 | 92.71 | 95.88 | 96.99 | 78.10 | 85.47 | 89.36 | 82.60 | 89.16 | 93.03 |
| 1 | | ✓ | 38.45 | 57.20 | 68.65 | 94.95 | 97.40 | 97.60 | 81.05 | 88.30 | 92.95 | 85.1 | 92.6 | 95.0 |
| 1 | ✓ | | 38.05 | 57.00 | 67.95 | 93.85 | 96.45 | 97.25 | 80.10 | 85.95 | 91.55 | 85.3 | 93.1 | 95.0 |
| 1 | ✓ | ✓ | 38.30 | 56.55 | 68.20 | 93.85 | 96.60 | 97.30 | 78.20 | 85.50 | 91.45 | 85.2 | 92.7 | 95.1 |
| 5 | | ✓ | 38.75 | 57.95 | 69.90 | 95.15 | 97.40 | 97.50 | 78.45 | 88.60 | 93.05 | **85.8** | 93.5 | **95.2** |
| 5 | ✓ | | 38.60 | 57.20 | 68.05 | 93.80 | 96.30 | 97.15 | 79.45 | 86.10 | 91.80 | 85.4 | **93.9** | 95.1 |
| 5 | ✓ | ✓ | 38.45 | 57.70 | 68.80 | 94.50 | 96.95 | 97.35 | 79.35 | 87.45 | 91.80 | **85.8** | 93.7 | 95.1 |
| 10 | | ✓ | 39.20 | 58.10 | 69.45 | 95.35 | 97.45 | 97.45 | 78.75 | 89.20 | 92.80 | 85.4 | 93.4 | **95.2** |
| 10 | ✓ | | 38.00 | 57.00 | 67.80 | 93.60 | 96.10 | 97.15 | 79.70 | 88.40 | 92.60 | 85.4 | 93.8 | **95.2** |
| 10 | ✓ | ✓ | **40.10** | 58.50 | 69.65 | 95.00 | 97.20 | **97.65** | 80.8 | **89.12** | **93.33** | 85.4 | **93.9** | **95.2** |
| 20 | | ✓ | 38.35 | 58.20 | 69.10 | 95.15 | 97.40 | 97.55 | 79.45 | 89.10 | 93.05 | 85.3 | 93.3 | **95.2** |
| 20 | ✓ | | 38.05 | 57.15 | 68.15 | 93.35 | 95.95 | 97.05 | 79.45 | 86.10 | 91.80 | 85.7 | **93.9** | 95.0 |
| 20 | ✓ | ✓ | 39.45 | 58.50 | 69.65 | 95.20 | **97.55** | 97.55 | 78.40 | 87.40 | 93.25 | 85.2 | 93.7 | 95.1 |
| det. | | ✓ | 37.60 | **59.40** | **71.20** | **95.40** | 97.50 | 97.30 | 79.90 | 89.10 | 92.80 | 85.6 | 93.5 | 95.1 |
| det. | ✓ | | 38.40 | 58.30 | 70.10 | 92.00 | 95.10 | 96.60 | 78.70 | 88.70 | 92.90 | 85.3 | 93.7 | **95.2** |
| det. | ✓ | ✓ | 37.60 | 58.90 | 71.00 | 95.20 | 97.50 | 97.50 | **81.30** | 88.80 | 92.50 | 85.3 | 93.5 | **95.2** |

## 4.3 QUALITATIVE EVALUATION

Figure 6 illustrates the selection of instances and the course of FALCUN's relevance scores $r(x)$ over one acquisition round on a 3-class MNIST task (also used for the visualisation in Figure 1) for better interpretability. Yellow regions indicate a high score promoting regions of high interest. Initially, all instances with high uncertainty, primarily located at the decision boundary, receive higher scores (see Figure 6a). The score in the surrounding of the selected instance (red circle) gets darker as the objects located close to it receive a smaller diversity score (see Figure 6b). In the first iterations, uncertain, but still diverse instances are preferred. In Figure 6d we derive a diverse set located in all three clusters mainly consisting of objects from uncertain areas. In Figure 7, we analyze FALCUN's selection on Openml-156 (Figure 7a) and BloodMNIST (Figure 7b). It effectively finds instances majorly located in regions where the classifier has more confusion (darker areas) while still enhancing diversity and not oversampling certain regions. E.g., on the right, most instances are chosen from the two most uncertain classes ($\sim 55\%$ accuracy). In contrast, only two objects are selected from the most confident class where the model already achieves $\sim 99\%$ accuracy.

## 4.4 ABLATION

Table 3 shows the influence of individual components of FALCUN for representative datasets from each investigated type of data. We vary $\gamma$, where smaller values lean towards uniform selection and larger values lean towards deterministic selection, including a completely deterministic selection (det.). The worst results (red values) are always reported for a complete random selection ($\gamma = 0$, top row) or a complete deterministic selection (bottom rows). We note that plain uncertainty sampling (second last row) is not suitable for the highly redundant RepeatedMNIST. Combining uncertainty and diversity for the relevance score is mostly better than only focusing on one aspect for a specific $\gamma$ value. Overall, our default value $\gamma = 10$ yields very good results on all datasets. Note that FALCUN's performance is largely not influenced by the exact choice for $\gamma$ in the range of 5 to 20.

## 5 CONCLUSION

We introduced FALCUN, a novel deep AL method that employs a natural transition from emphasizing uncertain instances at the decision boundary towards enhancing more batch diversity. This natural balance ensures robust label efficiency on varying datasets, query sizes, and architectures, even on highly redundant datasets. Moreover, FALCUN only operates on the output probability vectors, denoting faster acquisition times than many established methods that perform a thorough search through the high-dimensional embedding space of a neural network.

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

# A   FURTHER BACKGROUND AND EXPLANATIONS OF FALCUN

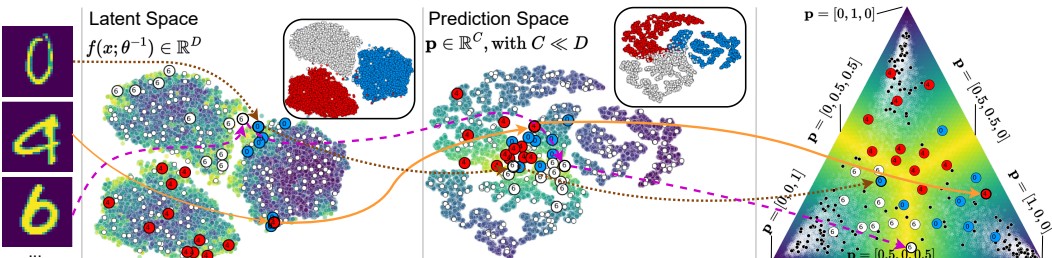

Figure 8: FALCUN selects diverse and uncertain instances (colored circles) in the probability space (see 3-class simplex on the right). In the latent space on the left, they cover the most informative regions (yellow) while being highly diverse and stemming from different clusters.

FALCUN harmoniously leverages the probability space for an efficient, diverse, and uncertain selection. Consider the three-class classification problem shown in Figure 8 as an example. The original data inputs (left) are forwarded through the network. The second and third columns visualize the latent and the probability space in a 2D t-SNE visualization. The colors indicate uncertainty, with yellow, lighter regions indicating higher uncertainty. On the right, the 3-dimensional simplex $S$ is given by $S = \{(p_1, p_2, p_3)|p_i \geq 0, p_1 + p_2 + p_3 = 1\}$, where $p_1, p_2, p_3$ denote the posterior probability for classes 1, 2, and 3, respectively. The corners indicate a high probability for a certain class reflected by a darker color. The opposite side corresponds to zero probability for this class and uniform distribution for the other two classes. The center corresponds to a uniform posterior distribution over all classes. Small black and white dots indicate objects in $\mathcal{L}$ and $\mathcal{U}$, respectively. Larger blue, red, and white circles indicate instances selected by FALCUN: they are prevalently in very informative regions in the latent space while being highly diverse.

## A.1   FROM DETERMINISTIC TO UNIFORM SELECTION

Figure 9 shows the selection probabilities of points depending on their relevance scores for different values for $\gamma$. While small values for $\gamma$ lean toward a uniform selection, larger values for $\gamma$ approximate a deterministic selection of the most relevant instance according to $r(x)$.

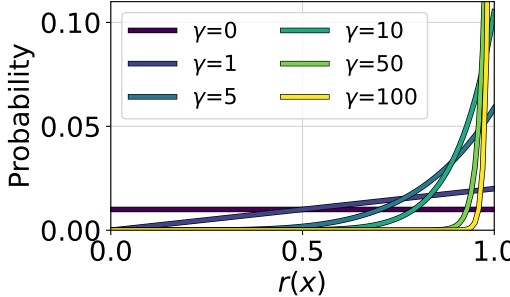

Figure 9: Selection probability of an instance $x$ for different $\gamma$ values as a function of its relevance score $r(x)$.

## A.2   CONNECTION OF UNCERTAINTY AND INITIAL DIVERSITY

In the following, we show the relation between margin uncertainty and distances to the one-hot encodings in the probability space.

Let $\hat{\mathbf{p}}$ be a one-hot encoding $\hat{\mathbf{p}}_i = \delta_i k$ of $\mathbf{p}(x)$ where $\delta_i k$ is the Kronecker Delta function and $k = \arg\max_c(\mathbf{p}(x))$ is the index of the most probable class. Further, let $\hat{\mathbf{p_2}}$ be a one-hot encoding

$\mathbf{p}(x)$ with $\hat{\mathbf{p_2}}_i = \delta_i k_2$ where $k_2 = \arg \max_2 (\mathbf{p}(x))$ is the index of the second most probable class.

$$d_{init}(x) = 1 - 1/2 \cdot |\underbrace{dist(\hat{\mathbf{p}}, \mathbf{p}(x))}_{\substack{\text{Distance to most} \\ \text{probable class}}} - \underbrace{dist(\hat{\mathbf{p_2}}, \mathbf{p}(x)))}_{\substack{\text{Distance to 2nd most} \\ \text{probable class}}}| \qquad (7)$$

$$= 1 + 1/2 \cdot \left( \sum_{i=1}^{C} |\hat{p}_i - p_i(x)| - \sum_{i=1}^{C} |\hat{p}_{2_i} - p_i(x)| \right)$$

$$= 1 + 1/2 \cdot \left( 1 - \mathbf{p}(x)[c_1] + \sum_{i=1}^{C} (i \neq k) p_i(x) - \left( 1 - \mathbf{p}(x)[c_2] + \sum_{i=1}^{C} (i \neq k_2) p_i(x) \right) \right)$$

$$= 1 + 1/2 \cdot (2 \cdot (1 - \mathbf{p}(x)[c_1]) - 2 \cdot (1 - \mathbf{p}(x)[c_2])) = 1 - (\mathbf{p}(x)[c_1] - \mathbf{p}(x)[c_2]) = u(x)$$

Therefore we initialize the diversity score with the uncertainty estimate: $d_{init}(x) := u(x)$.

# B EXPERIMENTAL DETAILS

## B.1 TRAINING AND ACTIVE LEARNING PARAMETERS

The implementation is in Python and uses PyTorch (Paszke et al., 2017), NumPy (Harris et al., 2020), and scikit-learn (Pedregosa et al., 2011). Our experiments have been evaluated on GPUs (NVIDIA GeForce RTX 2080 Ti) in an Ubuntu 20.04.2 LTS environment. For more details, we refer to our publicly available code base. An overview of the evaluated dataset and statistics is given in Table 4. BloodMNIST contains images from different normal cells belonging to eight classes, and DermaMNIST consists of dermatoscopic images categorizing seven different diseases (Yang et al., 2023). Example images are shown in Figure 11. We rescale images from the medical datasets from 28x28 to 32x32 with nearest-neighbor interpolation.

## B.2 DATASET PROPERTIES

We give an overview of our real-world examples' properties in Table 4. We regard image data in gray as well as colored and tabular data.

Table 4: Data set properties: number of points $N$, number of classes $C$, and number of features $F$.

| Type | Data set | N | C | F |
|---|---|---|---|---|
| Image (Gray) | MNIST | 60,000 | 10 | 28x28 |
| | RepeatedMNIST | 60,000 | 10 | 28x28 |
| | FashionMNIST | 60,000 | 10 | 28x28 |
| | EMNIST | 131,600 | 47 | 28x28 |
| Image (Color) | SVHN | 73,257 | 10 | 32x32x3 |
| | BloodMNIST | 11,959 | 8 | 28x28x3 |
| | DermaMNIST | 7007 | 7 | 28x28x3 |
| Tabular | OpenML-6 | 16,000 | 26 | 17 |
| | OpenML-156 | 800,000 | 5 | 11 |
| | OpenML-155 | 829,201 | 10 | 11 |

## B.3 REDUNDANCY IN REAL WORLD DATA SETS

Figure 11 shows some of the medical images from the Blood MNIST and the DermaMNIST data sets. We can see that within a class, images can be very similar, s.t. their information is redundant. A good AL method should not select such redundant objects for labeling in order to optimize the learning process.

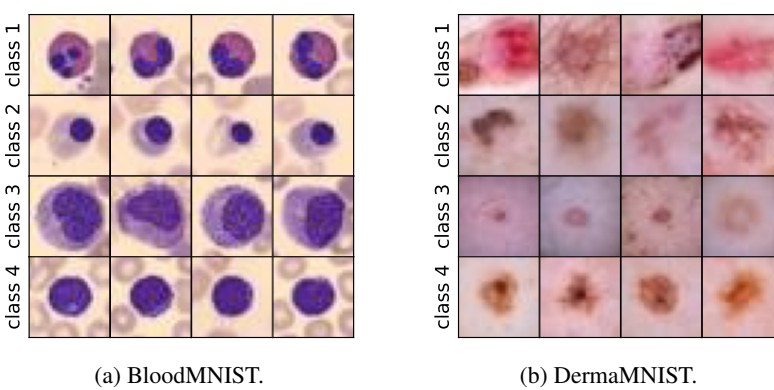

(a) BloodMNIST.          (b) DermaMNIST.

Figure 11: Exemplary images from the medical MNIST datasets.

# C  ALL LEARNING CURVES

In this Section, we report all learning curves for all tested datasets and settings, including grayscale data with varying query sizes (see Figure 13), RGB data with varying model architecture (see Figure 14), redundant data (see Figure 16), and tabular data (see Figure 17).

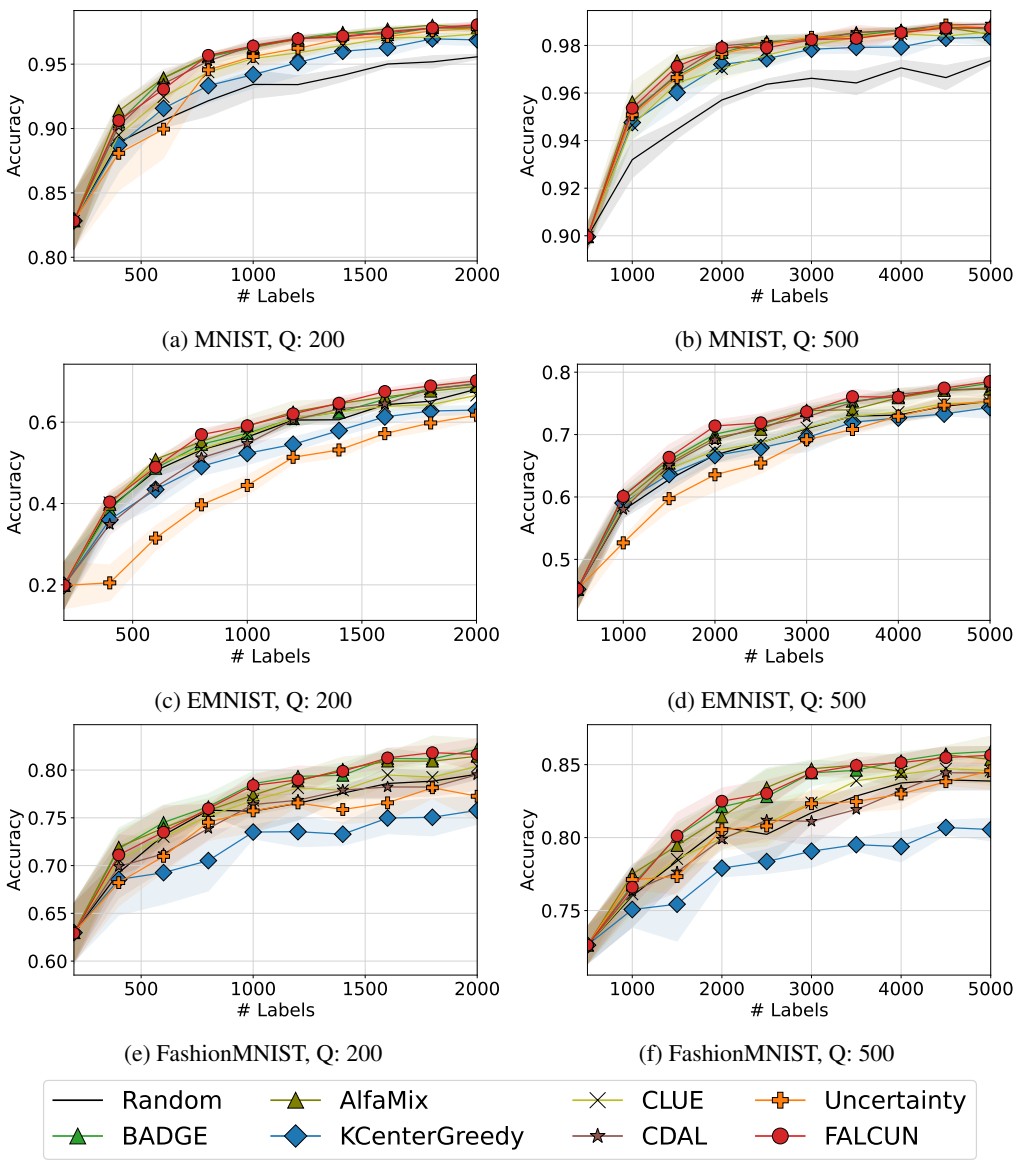

Figure 13: AL Curves grayscale images.

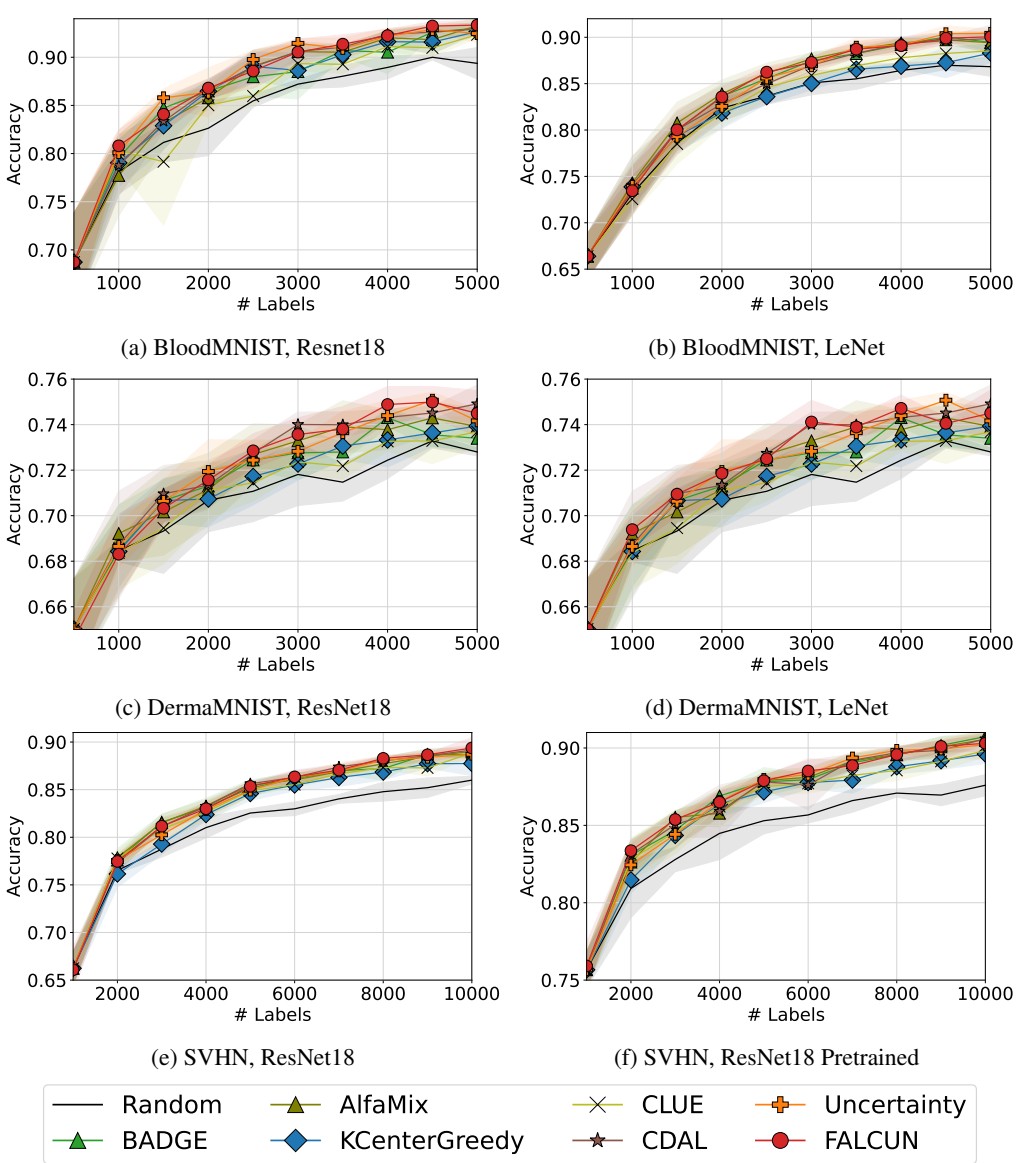

Figure 14: AL Curves colored image datasets.

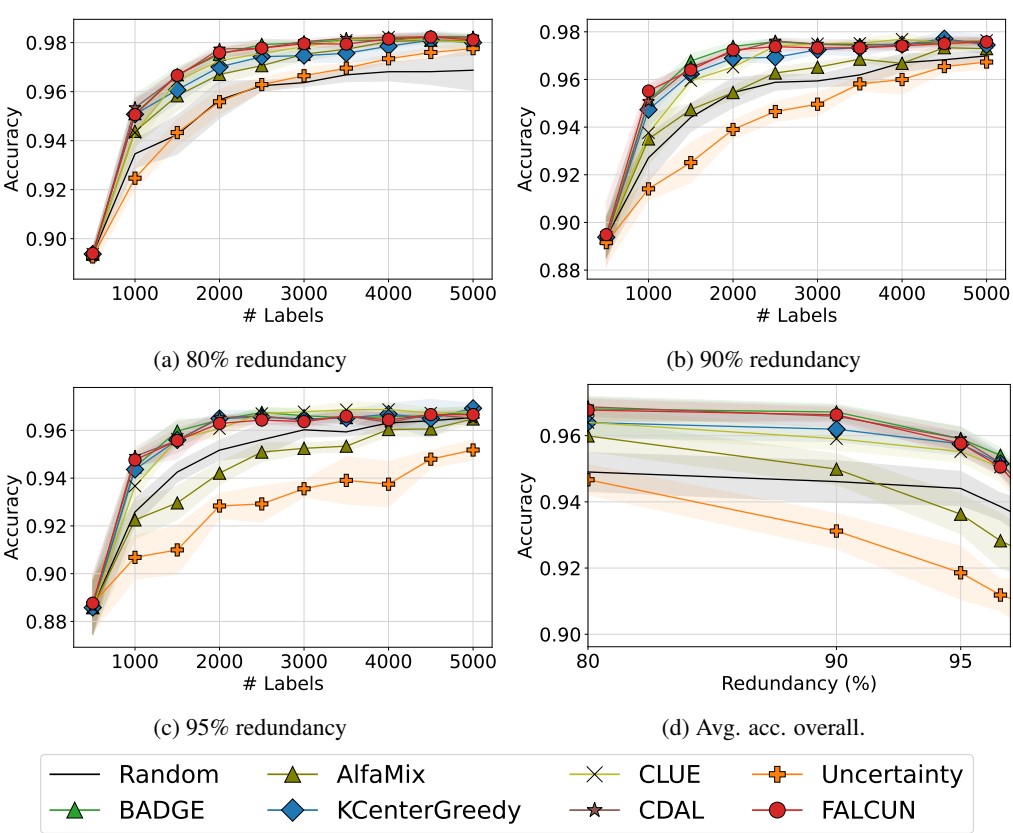

Figure 16: AL Curves on RepeatedMNIST.

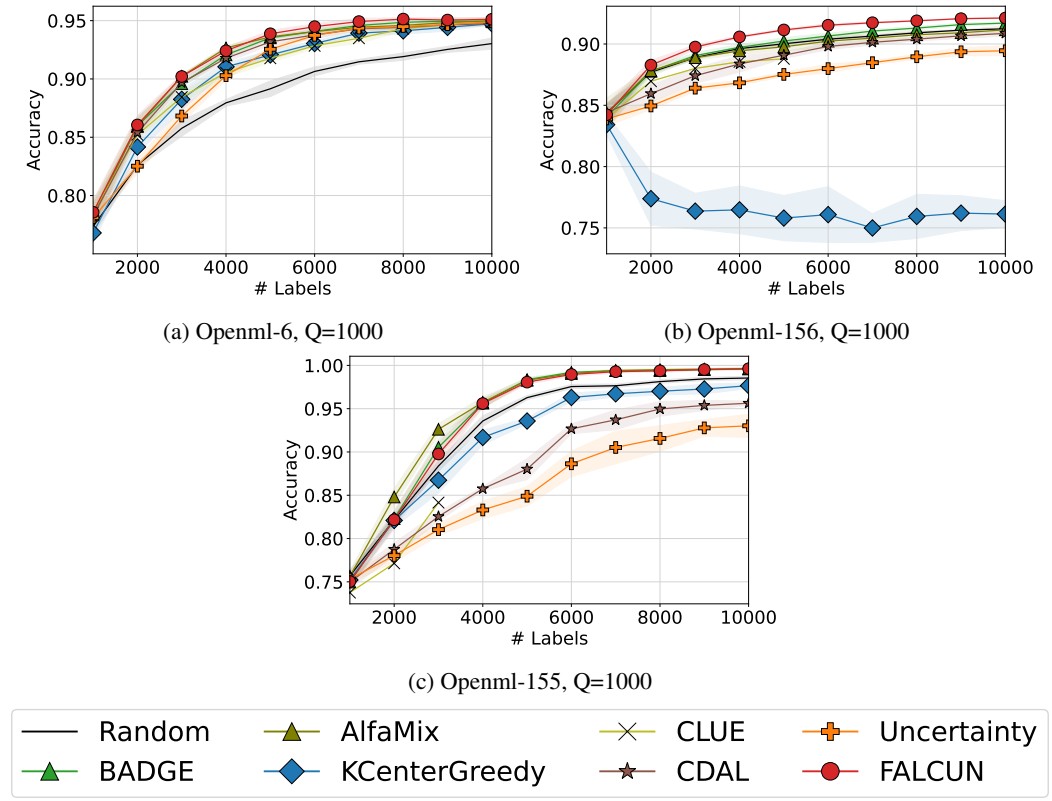

(a) Openml-6, Q=1000

(b) Openml-156, Q=1000

(c) Openml-155, Q=1000

Figure 17: AL Curves for tabular data.

# D  SCALABILITY

We perform a systematic scalability analysis using the largest of the evaluated datasets Openml-156, where we change the size of the unlabeled pool $N$, the query size $Q$ and the hidden dimensionality $D$ of the used network (see Figure 18). We stopped experiments after 10 days. The only methods exceeding this limit are BADGE and CLUE. FALCUN denotes fast and robust runtimes over varying settings.

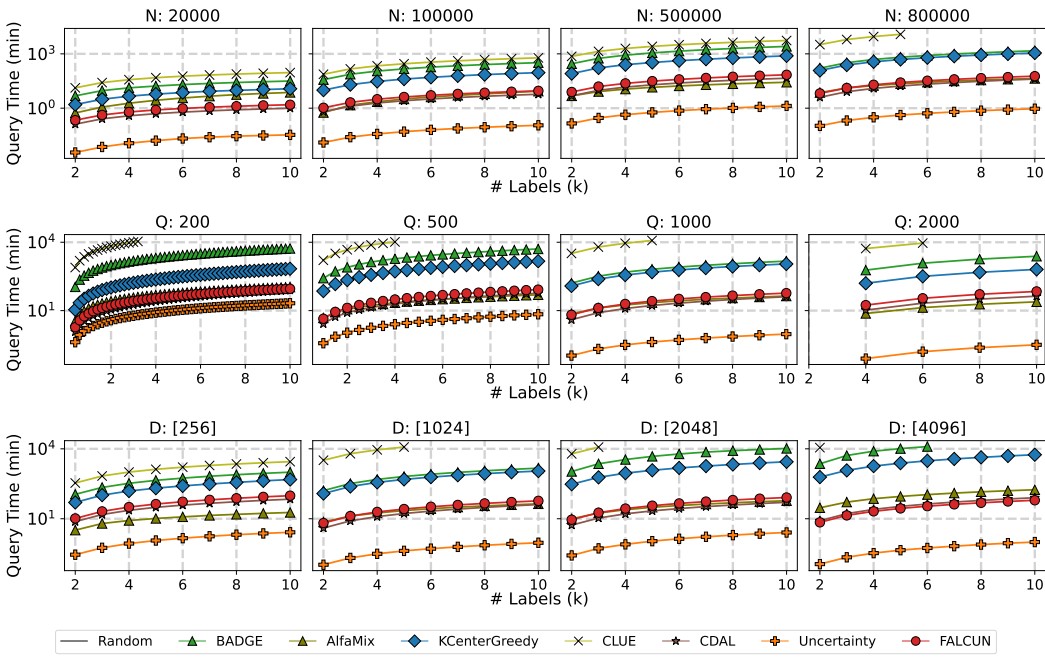

Figure 18: Average cumulated acquisition times (y-axis) on a log-scale vs. annotated samples (x-axis) over varying unlabeled pool sizes $N$ (first row), query sizes $Q$ (second row), and dimensionality of the penultimate layer $D$ (third row).

