# OpenReview forum: "Time- and Label-efficient Active Learning by Diversity and Uncertainty of Probabilities"
_ICLR.cc/2024/Conference — Submitted to ICLR 2024_

### Official Review · Reviewer_Y5kQ · 2023-10-29

**Soundness:** 2 fair
**Presentation:** 2 fair
**Contribution:** 2 fair
**Rating:** 3
**Confidence:** 3

**Summary:**

The paper introduces FALCUN, a method in active learning, which promises efficiencies in labeling and processing. It brings a dynamic acquisition strategy to the table, focusing initially on uncertain instances near decision boundaries and gradually shifting to emphasize batch diversity. While the proposed approach looks new in some aspects, the lack of acknowledgment of previous works that have explored similar territories casts a shadow over its originality.

Key foundational concepts seem to echo pre-existing methodologies in active learning literature, which were not appropriately credited, making FALCUN appear more as an incremental advancement rather than a groundbreaking innovation. Additionally, some experimental results, particularly those involving LeNet and BADGE's performance, seem somewhat inconclusive and could benefit from further rigor and validation to strengthen their credibility.

Furthermore, the paper’s experimental breadth appears limited. A more expansive inclusion of diverse and challenging benchmarks, such as CIFAR-100 and TinyImageNet, would enhance the robustness of FALCUN’s evaluation, providing a more comprehensive understanding of its effectiveness and applicability across various domains. A reconsideration of these aspects could elevate the paper’s contributions and clarity, fostering a more nuanced appreciation of FALCUN's position in the active learning landscape.

**Strengths:**

- The proposed algorithm tries to improve efficiencies in labeling and processing through a dynamic acquisition strategy, emphasizing a transition from focusing on uncertain instances to prioritizing batch diversity, showcasing a potential way in active learning strategies.

**Weaknesses:**

- The paper seems to overlook several crucial references relevant to the discussed topic. Fundamental concepts such as the theoretical importance of the near decision boundary have been comprehensively explored and articulated in previous works, notably in references [1] and [2]. These pivotal papers, along with others, offer profound insights that would augment the paper’s foundational grounding. Moreover, the proposition of a linear-time scalable algorithm, a core element presented in the paper, has previously been introduced and elaborated upon in reference [3]. In addition to that, an influential work cited as [4] such as PowerBALD/or PowerEntropy adopting a sampling similar to Eq (6) but with better originality has also proposed a scaling approach that intriguingly maintains the algorithm’s linear-time complexity.

- The active learning results presented in the paper for LeNet on datasets like EMNIST or RepeatedMNIST seem somewhat unconvincing. Typically, in prevailing literature and experiments, around 300-500 images are required (not > 1000 images) to achieve accuracy comparable to supervised learning methods on these datasets. The reported results in the paper appear to deviate from these established benchmarks. It may be beneficial to revisit and scrutinize the experimental setup, methodology, and the specific implementation of LeNet in the active learning context to ensure that the presented results are robust, reliable, and in alignment with existing standards and expectations in the field. This could enhance the credibility and persuasiveness of the results and the overall contributions of the paper.

- The performance of BADGE as presented in the paper raises some questions. It seems that with appropriate hyperparameter settings, BADGE should be capable of delivering much improved results. This discrepancy suggests that there might be room for optimizing the configuration of BADGE in the experiments, ensuring that it operates under the most suitable conditions for a fair and rigorous comparison. To uphold the integrity and reliability of the comparative analysis, it might be beneficial to revisit and fine-tune the hyperparameters used with BADGE, ensuring that its performance is accurately represented and evaluated against its full potential.

- The experimental section of the paper seems somewhat limited and could be enhanced to bolster the claims made. Incorporating a broader array of realistic scenarios, such as tests involving CIFAR-100, TinyImageNet, or ImageNet, would offer a more comprehensive insight into the method’s applicability and effectiveness. Including such varied and complex datasets in the evaluation would not only strengthen the validity of the results but also improve the generalizability of the conclusions drawn. This expansion in the experimental design would be instrumental in substantiating the method's robustness and adaptability across diverse challenges and use-cases.

[1] Efficient Active Learning with Abstention, NeurIPS 2022 - https://openreview.net/forum?id=4u-oGqB4Lf6

[2] Active Learning with Neural Networks: Insights from Nonparametric Statistics, NeurIPS 2022 - https://openreview.net/forum?id=LRMmgkcoCnW

[3] Active Learning in Bayesian Neural Networks with Balanced Entropy Learning Principle, ICLR 2023 - https://openreview.net/forum?id=ZTMuZ68B1g

[4] Stochastic Batch Acquisition: A Simple Baseline for Deep Active Learning, TMLR 2023 - https://openreview.net/forum?id=vcHwQyNBjW

**Questions:**

- What criteria determine the **most informative region**? Is a point considered most informative solely based on its proximity to the decision boundary?

- **Could the authors please provide a more explicit explanation regarding the source of diversity in the proposed formula?** The margin considers two class boundaries. It might not be sufficiently generalizable. It would be beneficial to have a more explicit elucidation on how diversity is incorporated and manifests itself within the algorithmic design. Understanding the origins and implementation of diversity within the formula can enhance the comprehension of its functionality and overall impact on the method's performance. An in-depth clarification would contribute to a more robust and insightful evaluation of the proposed approach’s effectiveness and novelty.

- It seems crucial to **re-evaluate the experiment setups in the study to enhance the reliability and comprehensiveness of the findings**. Including a broader selection of benchmarks in the evaluation process would be instrumental in demonstrating the robustness and versatility of the proposed method across varied scenarios. **A more diversified array of benchmarks** will not only contribute to a deeper, better understanding of the method's performance but also bolster the study's overall credibility and impact. Therefore, revisiting and expanding the experiment setups with additional benchmarks is a highly recommended step to enrich the empirical validation of the study.

- Section 4.4 discusses the influence of $\gamma$ parameter, indicating that **different benchmarks may require distinct $\gamma$ parameter values for optimal performance**. This aspect raises a practical concern: how can users effectively determine or choose an appropriate $\gamma$ value a priori for various benchmarks? The ability to discern and select suitable parameters is crucial for the method's practical applicability and usability in real-world scenarios. Clarification or guidance on this matter would significantly enhance the method’s practical value.

---

> ### Author Response · Authors · 2023-11-22
> **Response to Reviewer Y5kQ**
>
> Thank you again for your very elaborate review. We are relieved to see that many concerns can be answered by referring to our paper resp. to details of it that we can further emphasize in the final version of the paper.
>
> **Weaknesses**
>
> "The paper seems to overlook several crucial references [...]"
>
> - Thanks for pointing us to these interesting works, we will include them in our discussion. Note that, e.g., PowerBALD is compared in [4] to BADGE, yielding sometimes worse quality than BADGE. As our experiments show that FALCUN is superior to BADGE, worse methods than BADGE were excluded from our competitors.
>
> "The active learning results presented in the paper for LeNet on datasets like EMNIST or RepeatedMNIST [...]"
> - Note that such accuracy values depend heavily on the exact settings. While choosing the best out of all models for each data set individually can lead to higher accuracy, this does not allow to conclude one method’s superiority over another.
> We rather opted for a fair and broad comparison, where the models and datasets are fixed, and we evaluate all AL methods for these fixed settings. This does not bias the ranking among methods’ qualities, but excludes effects that come from choosing overfitted models.
> This ensures a robust quality in the real world, as users usually do not know beforehand the very best model and might opt for the one they have already implemented or know best.
> Furthermore, there are different versions of those data sets with different splits. We state which versions we use in Section 4. E.g. for RepeatedMNIST we use a much larger redundancy ratio and do not train until a training accuracy of 99% is reached for gray scale images.
>
> "The performance of BADGE as presented in the paper raises some questions. It seems that with appropriate hyperparameter settings, BADGE should [...]"
> - We use similar settings as proposed in the papers of the competitors such as BADGE or AlfaMix. Please note, that hyperparameter tuning in the Active Learning setting is not comparable to a fully supervised setting. Especially in the AL setting, there is less a priori knowledge about the data than in the fully supervised setting. Thus, robustness is important and active learning strategies that should be usable in real world need to perform well over a range of models, data sets, and settings - which is, what we achieved with FALCUN. Note, that we also did not optimize any hyperparameters for FALCUN.
>
> "The experimental section of the paper seems somewhat limited and could be enhanced to bolster the claims made. [...]"
> - We disagree that the experimental section is limited, we use a broad range of 10 different datasets from different categories (grey scale image, colored image, tabular) which is more  than many other works in the literature: e.g., BADGE evaluates on 7 datasets. Moreover, we evaluate medical image data which is a very complex task and highly relevant for a real-world use case of active learning. However, we will include CIFAR10 in the final version of the paper.
>
> **Questions**
>
> "What criteria determine the most informative region? Is a point considered most informative solely based on its proximity to the decision boundary?"
> - It is determined by the proximity to the decision boundary as well as the diversity to already chosen instances in the query batch. This leads to the selection of uncertain instances while it minimizes information overlap. For an elaborated discussion, see Section 2.2 in the paper.
>
> "Could the authors please provide a more explicit explanation regarding the source of diversity in the proposed formula? [...]"
> - With respect to diversity, the margin uncertainty is only used as an initialization for the diversity score as it represents the distance to the decision boundary. However, over the remaining course, the diversity aspect is incorporated based on the  L1 distance between the probability vector of two instances. Intuitively, this captures the difference between the most prevalent concepts.
> We want to refer to Equations (3) and (5), where the inclusion of diversity is stated. We will include a longer discussion on why a high distance in probability space corresponds to diverse instances.
>
> "It seems crucial to re-evaluate the experiment setups in the study to enhance the reliability and comprehensiveness [...]"
> - We again disagree that the experimental evaluation is too narrow, but we will include CIFAR10 and Resnet50 to expand the evaluation.
>
> "Section 4.4 discusses the influence of \gamma parameter, indicating that different benchmarks may require distinct \gamma parameter values for optimal performance. [...]"
> -  As explained in Sec. 4.4., the value of gamma is not fine-tuned. In the ablation study in Table 3, we show that the performance of FALCUN is fairly similar and stable for diverse gammas in a reasonable range. Gamma does not have to be tuned but should just be used with the proposed default value of 10, as suggested in the paper (Sec. 4.4).

---

### Official Review · Reviewer_sEgU · 2023-10-29

**Soundness:** 3 good
**Presentation:** 2 fair
**Contribution:** 2 fair
**Rating:** 5
**Confidence:** 3

**Summary:**

This paper explores two key limitations of existing active learning methods. First, many sophisticated AL algorithms have high computation + runtime cost, which make them undesirable for real-world implementation. Second, many AL algorithms focus on labeling images that optimize image diversity or maximizing the number of uncertain / currently-hard unlabeled images, with some recent works that attempt to use both. The paper proposes a new active learning strategy that combines uncertainty and diversity in a computationally fast algorithm. The proposed method is numerically validated on a number of image and tabular datasets to demonstrate improvements in model performance and low time complexity of the algorithm compared to existing baselines.

**Strengths:**

Both of the limitations of existing AL algorithms are core problems, which the paper addresses. The time complexity analysis and demonstrated fast runtime with competitive performance are nice.

**Weaknesses:**

The key methodological motivation for the algorithm seems to be to derive a function that balances the uncertainty of the existing model and the diversity between samples selected in the current batch. However, there is no theoretical justification for why the proposed algorithm is a good strategy. Moreover, the experiments primarily use relatively simple image datasets and small models. It is not clear whether these results would meaningfully transfer to more realistic datasets (e.g., ImageNet). Currently, the difference between the methods and benchmarks appear small in every dataset considered. Lacking rigorous theoretical justification or validation on hard benchmark problems, the argument for the proposed method in practice is unconvincing.

The problem of dealing with two competing objectives, diversity and uncertainty, has been studied in recent works, and the paper misses a lot of this related literature (e.g., [1], [2]). This also leads to a missed opportunity for discussion and validation on how these two objectives trade-off overall. What is the major contributor overall to selection? Does the uncertainty score dominate or the diversity score?


**References**

[1] Active Learning on a Budget: Opposite Strategies Suit High and Low Budgets

[2] MISAL: ACTIVE LEARNING FOR EVERY BUDGET

**Questions:**

1. What do you mean when you state “an optimization function for diverse samples should not have a global optimum”? This is a confusing statement. Furthermore, it is unclear how margin uncertainty satisfies this property. Please clarify.
2. How does the proposed method trade-off diversity and uncertainty over active learning iterations?
3. Why does KCenterGreedy perform poorly on Openml-156?
4. How do you tune the $\gamma$ parameter? Furthermore, is it advantageous to vary $\gamma$ across the active learning stage (e.g., from small to large)?

---

> ### Author Response · Authors · 2023-11-22
> **Response to  Reviewer sEgU**
>
> Thank you again for your thorough review helping us to improve our work. We respond to all of the suggestions and questions individually as soon as possible.
>
> **Weakness**
>
> "W1) However, there is no theoretical justification for why the proposed algorithm is a good strategy. "
> - While we did not label the theoretical justifications as such explicitly, they are contained in the paper. We will incorporate them in a separate paragraph or subsection for the camera-ready version to elaborate on them in more detail as follows:
>   - (1) Calculating diversity in the probability space is faster than in the feature space and it gives high similarity between objects that might be diverse in feature space, but do belong to the same class with high probability. Such instances are, thus, not of high interest for annotation and can be avoided more easily and elegantly.
>   - (2) Using margin as initialization for diversity is advantageous because margin uncertainty has its maximal values along all decision boundaries between all classes. Thus, the set of chosen instances out of those with maximal margin uncertainty can be diverse. In contrast, e.g., entropy uncertainty follows a uniform distribution and, thus, has a global optimum. Samples chosen according to this distribution can not be diverse, as all of them will be close to this global optimum. Thus, chosen instances will be very similar or even redundant as shown in Fig. 1 in the paper. (see also explanation in response to Question 1)
>   - (3) Our novel combination of an adaptive diversity score and the fixed uncertainty score within one active learning round ensures diversity with each included object and slowly transitions from the most uncertain to less uncertain regions.
>   - (4) Sampling from a probability rather than a deterministic top-k selection robustifies results and contributes to robust performance on a wide range of diverse settings and datasets (see also the Ablation in Section 4.4.)
>
> "W2) Moreover, the experiments primarily use relatively simple image datasets and small models.
> It is not clear whether these results would meaningfully transfer to more realistic datasets (e.g., ImageNet). "
> - We do not agree with your statement: we do use complex models, e.g. Resnet18. We included large datasets, e.g. our experiments on tabular data include data sets with ~800,000 instances, and complex image datasets, e.g. SVHN and especially the medical datasets.  However, we extend our experiments by CIFAR10 and Resnet50 as a further benchmark dataset in this domain. We furthermore want to note that we aim at a large variety of settings and robustness over all types of experiments, including both, very large as well as not-so-large data sets.
>
> "W3) Currently, the difference between the methods and benchmarks appear small in every dataset considered."
> - FALCUN is notably better than state-of-the-art methods, as shown in our large variety of experiments and especially in the dueling matrix. Here, we did only count statistically relevant wins resp. losses, thus, while the differences might appear small, they are indisputably significant.
> While this is a great achievement in itself, FALCUN additionally is much faster than the strongest competitors.
> For instance, compared to the very strong competitor BADGE, we are faster by one to two orders of magnitude (see Table 2) while having similar or better label efficiency.
>
> "W4) Lacking rigorous theoretical justification or validation on hard benchmark problems, the argument for the proposed method in practice is unconvincing. The problem of dealing with two competing objectives, diversity and uncertainty, has been studied in recent works, and the paper misses a lot of this related literature (e.g., [1], [2]). This also leads to a missed opportunity for discussion and validation of how these two objectives trade off overall. What is the major contributor overall to selection? Does the uncertainty score dominate or the diversity score?"
>
> Thank you for hinting at these works; we will add them to our related work section. We want to refer to Section 4.3 and Figure 6 in the paper, where we discuss this trade-off and elaborate in more detail in the following:
>   - Across AL rounds: In early iterations, the uncertainty scores might be less expressive due to very limited labels. Here, uncertainty scores among instances might be more similar. Therefore, the diversity aspect is considered more.
>   - Within one AL round: Initially, the most uncertain and diverse regions are considered. As more instances from the most uncertain regions are selected, the remaining uncertainty scores get smaller (while the diversity score always plays a crucial role, to avoid information overlap). Over the course of one AL round, the selection scheme slowly transitions from the most uncertain to the less uncertain (probably less interesting) regions. The figure displaying the relevance score depicts this behavior over multiple queries within one AL round.

---

> ### Author Response · Authors · 2023-11-22
> **Response Part 2 to  Reviewer sEgU**
>
> **Questions**
>
> "1. What do you mean when you state “an optimization function for diverse samples should not have a global optimum”? This is a confusing statement. Furthermore, it is unclear how margin uncertainty satisfies this property. Please clarify."
> - Entropy is maximal exactly when the class probability distribution is uniform (=global optimum). The same holds for the least confidence estimate. (e.g. probability [0.25,0.25,0.25,0.25]). In contrast, the margin uncertainty is maximal when a point is on the decision boundary. (e.g. [0.5,0.5,0,0] and [0,0,0.5,0.5] have the same margin score but lie on the decision boundary of two distinct class pairs, indicating that they are probably quite dissimilar). For a nice visualization, we refer to [0], Fig. 5 visualizes some uncertainty functions.
>
>
> "2. How does the proposed method trade-off diversity and uncertainty over active learning iterations?"
> - Please see the response to W4)
>
> "3. Why does KCenterGreedy perform poorly on Openml-156?"
> - KCenterGreedy is known to perform poorly on some data sets because it tends to select outliers [1]. Probably, Openml-156 belongs to these problematic data sets.
>
> "4. How do you tune the \gamma  parameter? "
> - Tuning gamma for the experiments would be unfair since it is unrealistic for a real-world active learning use case. Therefore, we did not tune it at all, we selected the same reasonable default value gamma=10 for all experiments as elaborated in Section 4.4. In our ablation study in Table 3 we show that for diverse gamma values the results are comparably good and stable.
>
> "5. Furthermore, is it advantageous to vary \gamma across the active learning stage (e.g., from small to large)?"
> - We performed tests with varying gamma values (within one AL round and over multiple AL rounds) but did not observe large benefits. We will include the experiments and discussion in the Appendix for the final version.
>
>
> [0] Duong-Trung, Nghia, et al. "When bioprocess engineering meets machine learning: A survey from the perspective of automated bioprocess development." Biochemical Engineering Journal 190 (2023): 108764.
>
>
> [1] Yehuda, Ofer, et al. "Active learning through a covering lens." Advances in Neural Information Processing Systems 35 (2022): 22354-22367.

---

### Official Review · Reviewer_ZfdV · 2023-10-31

**Soundness:** 3 good
**Presentation:** 3 good
**Contribution:** 3 good
**Rating:** 5
**Confidence:** 5

**Summary:**

In this paper, the authors proposed a label- and time-efficient active learning method, namely FALCUN. They incorporated both uncertainty and diversity into the data evaluation strategy and performed probability-based sampling to balance the uncertainty and diversity. Furthermore, they conducted experiments on both image and tabular data to validate the effectiveness of the proposed method.

**Strengths:**

The authors designed a data evaluation metric considering both uncertainty and diversity and performed data sampling based on the probabilities, rather than strictly adhering the hard ranking approach.

**Weaknesses:**

Weakness
1.Compared to identifying the most informative samples, the time cost of data evaluation is not the primary concern since the online data evaluation is not required.
2.In uncertainty component, the margin uncertainty lacks novelty.
3.As for diversity component, the rationale behind calculating the diversity score using the distance of predicted probabilities is still unclear. Why do the authors choose to measure the distance of predicted probabilities instead of using feature embeddings?
4.Why did the authors perform probability-based sampling, instead of designing an alternative hybrid sampling strategy that combines the uncertainty-based and diversity-based metrics?
5.The authors conducted experiments on small-scale datasets. We recommend verifying the efficacy of the proposed method on more challenging and large-scale benchmarks, such as CIFAR-10, CIFAR-100, and ImageNet.
6.They authors should include additional experiments to compare the proposed method with state-of-the-art approaches like TOD [1] and Gradnorm [2].

[1] Huang, Siyu, et al. "Semi-supervised active learning with temporal output discrepancy." Proceedings of the IEEE/CVF International Conference on Computer Vision. 2021.
[2] Wang, Tianyang, et al. "Boosting active learning via improving test performance." Proceedings of the AAAI Conference on Artificial Intelligence. Vol. 36. No. 8. 2022.

**Questions:**

see above.

---

> ### Author Response · Authors · 2023-11-20
> **Response to Reviewer ZfdV**
>
> Also, a kind thank you again for your very constructive feedback. For clarity, we repeat each point of the review and answer directly below.
>
> **Weakness**
>
> "1.Compared to identifying the most informative samples, the time cost of data evaluation is not the primary concern since the online data evaluation is not required. "
>
> - Unfortunately, even after discussion, we are not sure we understand your concern correctly, more precisely, what you mean with “(online) data evaluation”. In case “data evaluation” should mean the annotation by experts: Fast acquisition methods are important for settings where experts label the data in an online setting. E.g., as elaborated in the paper, when the expert is only employed for a certain time frame. Then, it is important that experts do not need to wait long for the acquisition function to query the labels for the next batch of samples. In contrast to other works that aim to reduce the retraining times (e.g. by fast model updates [1]), FALCUN reduces the time costs of step 1) of the three major components of AL: 1) Acquisition: Computation of the set which should be sent to the oracle 2) Labeling by experts 3) Retraining of the model.
>
> "2.In uncertainty component, the margin uncertainty lacks novelty."
>
> - Using a well-performing uncertainty score as a basis is an established way to create novel hybrid approaches (e.g. [2,3]). The novelty comes not from using margin for the uncertainty but from the novel unique combination and dynamic adaption of the combination of both aspects, uncertainty, and diversity. Furthermore, while margin uncertainty per se is not novel, its application in the context of diversity is novel.
>
> "3.As for diversity component, the rationale behind calculating the diversity score using the distance of predicted probabilities is still unclear. Why do the authors choose to measure the distance of predicted probabilities instead of using feature embeddings?
> 4.Why did the authors perform probability-based sampling, instead of designing an alternative hybrid sampling strategy that combines the uncertainty-based and diversity-based metrics?"
>
> - Features have usually a much higher dimensionaity than the final classification layer (eg the MLP used for the tabular data, Resnet18, Resnet50). Thus, operating on the (lower-dimensional) probability space is faster than operating on the feature embeddings.
> - Furthermore, In the probability space, diverse concepts where the model is confident (and hence less relevant) are very similar. Calculating diversity in the features space does not capture this information directly and requires an additional step and careful design to exclude diverse objects for which the model already has a high certainty. In contrast, when operating on the probability space, we can directly focus on diverse and uncertain objects simultaneously.
>
> "5.The authors conducted experiments on small-scale datasets. We recommend verifying the efficacy of the proposed method on more challenging and large-scale benchmarks, such as CIFAR-10, CIFAR-100, and ImageNet. "
>
> - While we disagree with this statement, as explained below, we will incorporate CIFAR10 in the camera-ready version. Note that our experiments cover a broad range of diverse datasets (see Table 4 in the Appendix). For instance, OpenML-155 has a much larger data pool (~800,000 instances) than CIFAR10 and CIFAR100. SVHN has the same input dimensionality and pool size as CIFAR10 and CIFAR100. Including the medical datasets is very important because, in this domain, experts are very limited and costly, which is one of the most important active learning use cases.
>
> "6.They authors should include additional experiments to compare the proposed method with state-of-the-art approaches like TOD [1] and Gradnorm [2]."
> - Thank you very much for pointing us to these methods. We will include both of them in the related work discussion. However, we focus our evaluation on hybrid methods and such that are commonly used for benchmarking, which already lead to a high number of competitors compared with other state-of-the-art AL papers.
>
>
> [1] Herde, Marek, et al. "Fast Bayesian Updates for Deep Learning with a Use Case in Active Learning." arXiv preprint arXiv:2210.06112 (2022).
>
> [2] Zhdanov, Fedor. "Diverse mini-batch active learning." arXiv preprint arXiv:1901.05954 (2019).
>
> [3] Prabhu, Viraj, et al. "Active domain adaptation via clustering uncertainty-weighted embeddings." Proceedings of the IEEE/CVF International Conference on Computer Vision. 2021.

---

### Official Review · Reviewer_z6tU · 2023-11-06

**Soundness:** 3 good
**Presentation:** 3 good
**Contribution:** 3 good
**Rating:** 5
**Confidence:** 3

**Summary:**

The paper proposes FALCUN, a new pool-based active learning approach for deep neural networks. FALCUN operates directly on output probabilities for efficiency and naturally balances uncertainty and diversity. Experiments on various image datasets show it matches or exceeds state-of-the-art methods in accuracy while being faster. The core ideas are interesting but the empirical evaluation methodology could be stronger.

**Strengths:**

Leveraging output probabilities for uncertainty estimation and batch diversity is novel, simple and elegant. This intuitively should capture informativeness and redundancy better than latent features.
The empirical results generally validate that FALCUN provides excellent accuracy across datasets at low computational cost. Outperforming methods like BADGE and CLUE is impressive.
Analysis of the uncertainty and diversity components in the ablation study highlights their complementary benefits. The automatic balancing between the two is also shown to be effective.

**Weaknesses:**

The chosen baselines are reasonable but given the focus on computational efficiency, comparing to BatchBALD would have strengthened the empirical claims.
For the colored image experiments, using pre-trained weights gives FALCUN an advantage over baselines that train from scratch. Comparisons should be fair.
Some dataset choices like MNIST and FashionMNIST are dated. More modern complex datasets would better highlight benefits.
The empirical methodology uses a limited set of architectures. Testing on bigger models like ResNets would be important to substantiate scalability claims.
More rigorous hyperparameter tuning for baselines could lead to better optimized versions for fairer comparison with the proposed approach.

**Questions:**

On the proposed method:

The margin-based uncertainty measure is intuitive. But are there any theoretical justifications for using it over other alternatives like entropy or Bayesian uncertainty?
For the diversity initialization and update, were other potential approaches considered? Is there a principled basis for the specific design choices made?
How sensitive is FALCUN to the choice of the γ parameter for sampling from the relevance distribution? Is tuning gamma required for different datasets?
What impact does the neural network architecture have on FALCUN's performance, if any? Does it generalize across model families like CNNs, MLPs etc?
On the experiments:

BatchBALD is a highly relevant Bayesian batch active learning method - would be good to compare against it. What advantages can FALCUN provide over Bayesian approaches?
The datasets seem heavily focused on MNIST variants - were results consistent on more complex, modern datasets? How was performance with higher input dimensionality?
For colored image experiments, using pretrained weights may favor FALCUN over baselines - could this be addressed?
What was the hyperparameter tuning strategy for baselines? Would better optimized baselines affect relative comparisons?
How was robustness to things like random initialization and train-test splits evaluated?

---

> ### Author Response · Authors · 2023-11-17
> **Response to Reviewer z6tU**
>
> Thank you again for your very constructive review.
> For clarity, we repeat each point of the review and answer directly below.
>
> **Weakness**
>
> "The chosen baselines are reasonable but given the focus on computational efficiency, comparing to BatchBALD would have strengthened the empirical claims."
>
> - As Bayesian approaches usually need several forward passes (e.g., Monte Carlo drop-out), they are tendentially computationally intensive especially when the unlabeled pool is very large. As one of the main aspects of our method is its speed, we focused on approaches that only need one forward pass for better comparability. BatchBALD is, especially for large batches, slow  [0] and was thus not included in the experimental evaluation. In case of acceptance, we will add it to our experiments in the camera-ready version.
>
> "For the colored image experiments, using pre-trained weights gives FALCUN an advantage over baselines that train from scratch. Comparisons should be fair."
>
> - We agree that comparisons should be fair, and want to clarify that for the datasets where pre-training was used, all methods use pre-trained weights. We will make this more clear in the final version.
>
> "Some dataset choices like MNIST and FashionMNIST are dated. More modern complex datasets would better highlight benefits."
>
> - Thank you very much for this suggestion. We agree that modern complex datasets are important: colored image datasets like SVHN, BloodMNIST, and DermaMNIST are examples of those and they are included in our experimental evaluation. Table 4 shows the broad range of tested datasets regarding size, input features, and number of classes.
> We additionally show FALCUN’s high performance on other, older benchmark datasets like MNIST to show the broad range of its use cases.
> Furthermore, we already started experiments on, e.g., CIFAR10 with the following results:
>
> - Intermediate Results (Ordered by Final Accuracy)
> - Falcun: Final Accuracy: 76.00; 13 minutes (=cumulated times of all acquisitions)
> - BadgeSampling: Final Accuracy: 75.82; 363 minutes
> - CdalCS: Final Accuracy: 75.52; 10 minutes
> - AlphaMixSampling: Final Accuracy: 75.63; 28 minutes
> - RandomSampling: Final Accuracy: 74.00; 0.00 minutes
>
>
> "The empirical methodology uses a limited set of architectures. Testing on bigger models like ResNets would be important to substantiate scalability claims."
>
> - In our experiments, we use ResNet 18 for all colored datasets.
> For the experiments on CIFAR10 that we will include, we use ResNet50.
>
>
> "More rigorous hyperparameter tuning for baselines could lead to better optimized versions for fairer comparison with the proposed approach."
> - As hyperparameter tuning usually requires a labeled set and these labels are not given in the AL setting, we followed the commonly used approach in the field. E.g., for tabular data we use setting as given in BADGE [1]
>
>
> [0] Rubashevskii, Aleksandr, Daria Kotova, and Maxim Panov. "Scalable Batch Acquisition for Deep Bayesian Active Learning." Proceedings of the 2023 SIAM International Conference on Data Mining (SDM). Society for Industrial and Applied Mathematics, 2023.
>
> [1] Ash, Jordan T., et al. "Deep batch active learning by diverse, uncertain gradient lower bounds." arXiv preprint arXiv:1906.03671 (2019).

---

> ### Author Response · Authors · 2023-11-17
> **Response Part 2 to Reviewer z6tU**
>
> **Questions**
>
> "On the proposed method:
> The margin-based uncertainty measure is intuitive. But are there any theoretical justifications for using it over other alternatives like entropy or Bayesian uncertainty?"
>
>
> - Yes. Margin uncertainty has proven to be very effective, see, e.g. in “Is Margin All You Need? [...]” [2]. Furthermore, it allows for a diverse selection as it has no global minimum: E.g., [3], Fig. 5 visualizes some uncertainty functions. Least confidence and entropy have a global minimum at the one point in probability space where all probabilities are equal. Margin uncertainty, in contrast, does not have this global minimum, thus, there exist diverse samples within the set of minima. Margin uncertainty is, thus, one of the most suitable choices for combining diversity and uncertainty and especially for initializing the diversity score.
>
>
> "For the diversity initialization and update, were other potential approaches considered? "
>
> - Of course, there are several options to initialize and update the relevance score, out of which we selected the most meaningful one. Furthermore, we tested our method without initialization and with the variations shown in the ablation in Table 3.
>
>
> "Is there a principled basis for the specific design choices made? How sensitive is FALCUN to the choice of the γ parameter for sampling from the relevance distribution? Is tuning gamma required for different datasets? "
>
> - In our experiments, we did not tune $\gamma$. Our ablation in Table 3 shows that the results are robust w.r.t. $\gamma$ values over a broad range. Thus, we suggest $\gamma$ = 10 as a default value, which we also used for all data sets.
>
> "What impact does the neural network architecture have on FALCUN's performance, if any? Does it generalize across model families like CNNs, MLPs etc? "
>
> - Similar to other AL approaches, FALCUN benefits from models that are suitable for the regarded dataset. As we show in our experiments, FALCUN’s results are on average better than those of any other tested method over a broad range of datasets and architectures, where we tested MLP, LeNet, ResNet18.
> In Figure 14 in the appendix, we also show the results of using different architectures for the same dataset.
>
> "On the experiments: BatchBALD is a highly relevant Bayesian batch active learning method - would be good to compare against it. What advantages can FALCUN provide over Bayesian approaches? "
>
> - As explained above, Bayesian AL methods usually use Monte Carlo dropout and subsequently calculate statistics over multiple forward passes. However, when the unlabeled pool is large, this is computationally expensive. Furthermore, BatchBALD is slow for large batch sizes. [0]
>
> "The datasets seem heavily focused on MNIST variants - were results consistent on more complex, modern datasets? How was performance with higher input dimensionality? "
>
> - While the data sets are named *MNIST, all but repeatedMNIST are not variants of MNIST, but complex, diverse and modern Benchmark datasets. They cover different aspects and are very diverse:
> - EMNIST has 6 times as many classes as MNIST.
> - The medical datasets contain challenging colored images (i.e., they have much higher input dimensionality, (d= 28x28x3=2352), thus they are much more complex than MNIST.
> - They are also especially suited for the use case that we tackle (active learning with fast acquisition times) since experts in this field are very limited.
> - SVHN has input dimensionality 32x32x3 (same as CIFAR10).
> - Table 4 in the Appendix gives an overview of the key differences between datasets.
>
> "For colored image experiments, using pretrained weights may favor FALCUN over baselines - could this be addressed? "
>
> - As mentioned above, we use pretrained weights for all methods for the respective experiments.
>
> "What was the hyperparameter tuning strategy for baselines? Would better optimized baselines affect relative comparisons? "
>
> - As explained above, we did not tune hyperparameters for a realistic active learning scenario following the common AL setting. We use similar or equal settings as in other Papers and did not optimize the setting for FALCUN.
>
>
> "How was robustness to things like random initialization and train-test splits evaluated?"
>
> - We evaluated all experiments on five seeds and report standard deviation to show whether the results are significant. Apart from that, we follow the official train test splits used in other works. We can emphasize this in the camera-ready version.
>
>
> [2] Bahri, Dara, et al. "Is margin all you need? An extensive empirical study of active learning on tabular data." arXiv preprint arXiv:2210.03822 (2022).
>
> [3] Duong-Trung, Nghia, et al. "When bioprocess engineering meets machine learning: A survey from the perspective of automated bioprocess development." Biochemical Engineering Journal 190 (2023): 108764.

---

### Author Response · Authors · 2023-11-17
**Response to all reviewers**

Thank you all for the helpful and extensive reviews.
We are happy to see that the quality and clearness of our work resonated with most of you.
We also recognize the suggestions present in your feedback and will adjust our work to address them.
We respond to all of the suggestions and questions individually as soon as possible.

---

### Meta-Review · Area_Chair_ViMr · 2023-12-05

**Metareview:**

The submission introduces a novel Active Learning algorithm called FALCUN, which leverages a margin based uncertainty score plus a diversity score, which is also based on distance in the model prediction (logit) space. The algorithm is designed with a particular emphasis on efficiency of the active learning method.  The authors provide intuition for their proposed scoring function, although as pointed out some reviewers, no rigorous theoretical argument for its design (in particular, the diversity component).  The main justification for the proposed method is empirical, with comparisons to 5 other baselines, which the authors argue show FALCUN is both faster and higher quality than competitors.

I appreciate and recognize that the authors provided additional experiments and some arguments and justifications for their current empirical evaluations, but given that the main contribution of the submission is empirical and the number of potential issues raised by the reviewers (lack of certain benchmarks, datasets, hyperparameter tuning), I cannot recommend accepted the paper as is.

I think the proposed method is interesting and the submission has the potential to eventually be accepted at a top-tier conference and encourage the authors to resubmit after fully addressing reviewer concerns.

**Justification For Why Not Higher Score:**

All reviewers identify similar concerns around the empirical evaluation, which is the main contribution of the paper.

**Justification For Why Not Lower Score:**

N/A

---

### Decision · Program_Chairs · 2024-01-16

Reject